# Tomato Bio-Protection Induced by *Pseudomonas fluorescens* N21.4 Involves ROS Scavenging Enzymes and PRs, without Compromising Plant Growth

**DOI:** 10.3390/plants10020331

**Published:** 2021-02-09

**Authors:** Ana García-Villaraco, Lamia Boukerma, Jose Antonio Lucas, Francisco Javier Gutierrez-Mañero, Beatriz Ramos-Solano

**Affiliations:** 1Facultad de Farmacia, Universidad San Pablo-CEU Universities, P.O. Box 67, Boadilla del Monte, 28668 Madrid, Spain; anabec.fcex@ceu.es (A.G.-V.); boukermalamia@gmail.com (L.B.); alucgar@ceu.es (J.A.L.); jgutierrez.fcex@ceu.es (F.J.G.-M.); 2Laboratoire National de Recherche en Ressources Génétiques et Biotechnologies, ENSA (ES1603), Al Harrach 16131, Algeria; 3Laboratoire de Protection et de Valorisation de Ressources Agro-Biologiques, Faculté SNV, Université Saad Dahleb Blida 1, Blida 09000, Algeria

**Keywords:** induced systemic resistance (ISR), *Pseudomonas fluorescens*, photosynthesis, pathogen related proteins (PRs), ROS scavenging cycle, SOD, priming, bio-protection

## Abstract

Aims: to discover the interrelationship between growth, protection and photosynthesis induced by *Pseudomonas fluorescens* N21.4 in tomato (*Lycopersicum sculentum*) challenged with the leaf pathogen *Xanthomonas campestris*, and to define its priming fingerprint. Methods: Photosynthesis was determined by fluorescence; plant protection was evaluated by relative disease incidence, enzyme activities by specific colorimetric assays and gene expression by qPCR. Changes in Reactive Oxygen Species (ROS) scavenging cycle enzymes and pathogenesis related protein activity and expression were determined as metabolic and genetic markers of induction of systemic resistance. Results: N21.4 significantly protected plants and increased dry weight. Growth increase is supported by significant increases in photochemical quenching together with significant decreases in energy dissipation (Non-Photochemical Quenching, NPQ). Protection was associated with changes in ROS scavenging cycle enzymes, which were significantly increased on N21.4 + pathogen challenged plants, supporting the priming effect. Superoxide Dismutase (SOD) was a good indicator of biotic stress, showing similar levels in pathogen- and N21.4-treated plants. Similarly, the activity of defense-related enzymes, ß-1,3-glucanase and chitinase significantly increased in post-pathogen challenge state; changes in gene expression were not coupled to activity. Conclusions: protection does not compromise plant growth; N21.4 priming fingerprint is defined by enhanced photochemical quenching and decreased energy dissipation, enhanced chlorophylls, primed ROS scavenging cycle enzyme activity, and glucanase and chitinase activity.

## 1. Introduction

Photosynthesis is the main process controlling plant growth. However, a number of stress factors affect this process and plants modulate it to ensure survival [1]. Among the challenges that plants overcome throughout their life are pathogen threats. Plant pathogens usually seek plants as a source of C skeletons obtained from photosynthesis, so they have developed the mechanisms to control this process to ensure C supply [2]. In turn, plants have developed mechanisms to overcome pathogen takeover, redirecting C sources to synthesis of defense compounds, which usually results in a fitness cost, compromising growth to ensure survival [3]. Hence, photosynthetic activity plays a pivotal role connecting redox status of the cell with carbon metabolism, growth and signal transduction pathways [1], which start within seconds from triggering stress and extend systemically through pulses of expression driven by Reactive Oxygen Species (ROS) waves [4].

In line with this reasoning, it is widely accepted that some non-pathogenic microorganisms are able to modulate plant defensive metabolism. Most studies on this topic deal with induction of plant protection triggered by pathogenic microorganisms, (Systemic Acquired Resistance, SAR), or by specific non-pathogenic microorganisms, (Induced Systemic Resistance, ISR), aiming to unravel signal transduction pathways, activation of gene expression involved in defense and identification of bacterial determinants and plant receptors involved therein [5]. The literature reports that ISR triggered by specific non-pathogenic microorganisms is a two-stage process in which defense metabolism is pre-stimulated, although this stimulation is not evidenced until stress challenge; this metabolic status is termed priming and may compromise plant growth [6]. Further studies have defined the priming phase as the period between the triggering stimuli and challenge, involving ROS changes together with undetectable genetic reprogramming, and the post-challenged priming state where protection is associated with synthesis of phytoalexins or defensive molecules [7,8]. Furthermore, although ROS generation is one of the earliest responses of plant to pathogen, as it is involved in the signaling pathways of defense reactions [9], ISR is also mediated by alteration of the redox status of cells, as it is involved in signal transduction [10,11,12].

Interestingly, alteration in the redox balance occurs in response to abiotic [13] and biotic stress [14]. Hence, the redox status of the cell connects plant protection, photosynthesis and plant fitness and productivity, an approach to plant-microbe interaction that is being explored in non-pathogenic strains. Hence, ROS scavenging cycle enzymes play a pivotal role in all stress situations. Tolerance to plant stress is achieved with superoxide dismutase (SOD), an enzyme in the first line of defense against the elevated levels of ROS. SOD removes O_2_^●^ by catalyzing its dismutation, one O_2_^●^ being reduced to H_2_O_2_ and another oxidized to O_2_ [15]; ascorbate peroxidase (APX) removes H_2_O_2_ and the system is regenerated in the ascorbate–glutathione cycle which is the main antioxidant pathway in plant cells linking the protection against ROS to redox-regulated plant defense [16]. Furthermore, upon stress, the production of highly oxidizing ROS immediately affects photosynthesis, even before symptoms are noticed on the leaves [17]. Therefore, plant stress can also be evaluated by measuring photosynthetic efficiency such as Fv/Fm (maximal Photosystem II -PSII- quantum yield), ΦPSII (effective PSII quantum yield) and NPQ (non-photochemical quenching) [17,18].

In summary, the events that take place in the plant at genetic, metabolic and physiological levels following a triggering stress have been defined by the term “priming fingerprint” [19] and seems to depend on the bacterial strain that triggers plant metabolism [20]. *Strain P. fluorescens* N21.4 has been extensively studied. Its capacity to trigger plant growth and protection against leaf pathogens in *Arabidopsis thaliana*, tomato and soybean has been shown [21,22,23], as well as its ability to trigger secondary metabolism in soybean and blackberry [24,25]. Furthermore, isolation of putative metabolites responsible to trigger a physiological response have been identified [26] and the metabolic pathway involved in the defensive response has been envisaged [12]. However, the metabolic events underlying the systemic response in terms of ISR and antioxidant metabolism have not been studied in detail.

Therefore, the aim of this study was to define the priming fingerprint of *P. fluorescens* N21.4 in tomato at the physiological, metabolic and genetic levels, evaluating (i) its ability to enhance growth, protect plants and alter photosynthetic parameters (physiological level), (ii) evaluating changes in photosynthetic pigments and ROS scavenging cycle enzymes and defense enzymes activity (metabolic level) and (iii) expression of genetic markers of ISR, in the post-challenged priming phase and comparing to the non-challenged plants in the priming phase at three time points after pathogen challenge.

## 2. Results

### 2.1. Physiological Parameters: Bio-Protection, Photosynthesis and Growth

Bio-protection induced by *P. fluorescens* N21.4 was determined in 6 weeks-old tomato plants. N21.4 provided significant disease reduction (40%). Disease suppression (Figure 1) was achieved with an evident reduction in spot diameter (data not shown).

Protection was associated with a significant increase in root (40%) and shoot (17%) dry weight of inoculated plants (Figure 2a). Enhancement of root dry weight was coupled with promotion of stem length either from the collar to the first leaf or to the last leaf (12%) (Figure 2b).

Chlorophylls and fluorescence parameters were evaluated one week after *X. campestris* challenge; results are presented in Figure 3. Maximal PSII quantum yield (Fv/Fm) ranged between 0.84 and 0.85 with no significant differences between treatments (Figure 3a). Photochemical quenching (PSII quantum yield = ΦPSII) was significantly increased by the strain (Figure 3b) and maintained after pathogen challenge, together with a minimum fluorescence value (Fo), similar to the healthy control, and lower than the pathogen control plants (Figure 3c). Non-photochemical quenching (NPQ) was significantly decreased in all bacterial treated plants as compared to healthy controls (Figure 3d).

### 2.2. Metabolic Parameters: Pigments and Enzyme Activities

Total chlorophylls (a + b) were significantly higher in N21.4-treated plants (Figure 4).

*P. fluorescens* strongly induced ROS scavenging enzyme activity in the pre- (control vs. N21.4 treated plants) and post-challenged (pathogen vs. pathogen + N21.4) states, changes being more marked in the post-challenged state (Figure 5). SOD (Figure 5a) and GPX (Figure 5d) behaved similarly. SOD activity was constant in healthy controls at the three time points; significant increases in SOD activity were recorded in the pre-challenged state in N21.4 treated plants as compared to control plants as well as in the post-challenged state compared to pathogen controls in the first sampling time (24 h); then, a decrease to basal values was registered at the second sampling time (48 h), to increase again at the third (72 h) sampling time; increases were similar when either the pathogen or N21.4 were used, although it was even higher on N21.4 + pathogen treated plants, being significantly different from the individual bacterial treatments. APX (Figure 5b) changed significantly only in the post-challenged state at 48 h on N21.4 + pathogen treated plants that declined to basal levels at the last sampling time. As regards GR activity (Figure 5c), values were constant at the three time points in all treatments except for pathogen challenged plants that decreased from 24 to 72 h; N21.4 treated plants showed higher values than controls (pre-challenged state) as well as in the post-challenged state, which showed the highest values. PPO activity (Figure 5e) showed the highest values on pathogen challenged plants, these differences being significant from all other treatments. In summary, all enzyme activities except PPO were highest in N21.4 + pathogen challenged plants.

The evaluated PR activities, ß-1,3-Glucanase (PR2) and chitinase (PR3) behaved similarly (Figure 6). Healthy controls and bacterial inoculated plants showed constant and similar levels at the three time points. However, while glucanase activity showed significantly higher levels in N21.4 + pathogen challenged plants at the three time points, chitinase activity increased only 48 h after pathogen challenge and remained high; hence, although there were no changes in the pre-pathogen challenge plants, significant increases were evidenced in the post-challenged state.

### 2.3. Genetic Markers

Changes in gene expression of PR1, PR2 and PR3 were not consistent with changes in enzyme activity (Figure 7); PR1 and PR3 showed a similar expression pattern, showing significant increases only on pathogen challenged controls 48 h after pathogen challenge. PR2 expression was similar in all treatments at 24 h but expression decreased significantly on pathogen challenged plants and remained at similar levels under N21.4 (pre and post-pathogen challenged state).

## 3. Discussion

The present study has demonstrated that *Pseudomonas fluorescens* N21.4 is able to enhance plant growth coupled to a higher photochemical quenching and diminishing energy dissipation, exerting a priming effect on ROS scavenging cycle enzymes as well as in PRs, and protecting the plant against *Xanthomonas campestris*; all these features conform its priming fingerprint.

Among PGPRs, fluorescent pseudomonads are appreciated for their contributions to increasing productivity of agricultural crops [27] by different mechanisms involving nutrient improvement and plant protection [5]. *P. fluorescens* N21.4 was able to protect tomato plants without compromising plant growth, consistent with other studies in different species [25,28]. Besides its direct role in basic production, photosynthesis also fuels and regulates a wide range of defense mechanisms, which may equally well affect plant productivity in terms of biomass and crop yield [1]. Photosynthesis represents energy input in the system, so our data supports optimization of light reactions by N21.4, based on the increase on photochemical quenching, which represents the energy funneled to C fixation, and the increase in the dissipated energy (Figure 3c,d), that may explain positive effects on dry weight (Figure 2). Interestingly, a low value of F0 (minimum fluorescence) has been reported as an indicator of low alert situation [17]; hence, the significantly higher values that appear in plants under biotic stress (Figure 3b), suggest F0 to be a good marker of biotic stress. On the other hand, N21.4 significantly increased the amount of total chlorophyll as compared to controls (Figure 4), consistent with the reported ability of *Pseudomonas* sp. to increase total chlorophyll level, which may also be related to the enhanced supply of certain nutrients such as nitrogen and phosphorus [29] and is consistent with the increase in dry weight and chlorophylls in the present study, also found in rice [17]. All these changes were detected in the post-challenge period.

As already mentioned, *P. fluorescens* N21.4 promoted plant growth (Figure 2) and was able to protect tomato plants against the leaf pathogen *Xanthomonas campestris* CECT 95 by 40–60% (Figure 1). This strain primed tomato plants as shown by the marked changes as compared to controls: strong increase in the activity of PRs, as well as activity of ROS scavenging enzymes’ activity upon pathogen challenge (Figure 6) [8,10,20]. Among enzymes of the ROS scavenging cycle and ascorbate-glutathione cycle, SOD, APX, GR and GPX (Figure 5) showed maximum values on N21.4 + pathogen challenged plants as compared to plants treated with only one strain, either N21.4 or the pathogen, therefore evidencing the more intense and quicker response upon stress challenge typical of priming [6,10]. However, polyphenol oxidase, which is involved in wound healing, pathogen defense, and several other cellular processes [30] was not among the primed enzymes (Figure 5e). PPO was highest on pathogen challenged controls, as expected for enzymes that are among the first line of protective enzymes in the hypersensitive response to block the pathogen progress, as in a typical SAR reaction [31]. On the other hand, the higher activity of SOD detected on pathogen treated plants is in accordance with the hypersensitive response typical of SAR, a basic mechanism in plant innate immunity. Interestingly, N21.4 treated plants showed similar levels of SOD activity as in pathogen controls, which speaks of the possibility of using SOD as a biochemical marker for biological stress on non-pathogenic strains. The common effect on SOD triggered by the two strains suggests the presence of common bacterial determinants that trigger plant metabolism, as plants and pathogens have been mutually overcoming their protective mechanisms in order to survive, so the various determinants may be differentially regulated [32]. Consistent with this statement, PGPRs have a variety of eliciting molecules to trigger ISR, either components of the bacterial cell surface or excreted metabolites [33,34,35]. In the case of N21.4, the effective protection triggered by the bacteria itself is also achieved by at least three different determinants (structural and metabolic) with similar levels of protection (data not shown). Further studies have shown that a metabolic fraction of N21.4 culture media is able to trigger the same response in Arabidopsis, that is, increasing ROS removing ability [28].

In addition to the priming effect detected on enzymes of the ascorbate-glutathione cycle, it should be kept in mind that ROS generation is one of the earliest responses of plant to pathogens [9] and is involved in the signaling pathways of defense reactions. The ascorbate–glutathione cycle serves as the main antioxidant pathway in plant cells linking the protection against ROS to redox-regulated plant defense [1,11]. Fluctuations detected on ROS scavenging enzymes activity are key to regulate quick adaptation to abiotic stress as it has been shown that signals integration occurs through pulses in gene expression modulated by ROS waves; the peaks in SOD 24 h after pathogen challenge (a.p.c.) and in APX 48 h a.p.c support this notion [4]. On the other hand, having an effective antioxidant network ensures high rates of photosynthesis [36], therefore improving ROS scavenging cycle activity will probably result in a better plant growth and yield [3].

Expression of the specific subset of PRs has been associated with SA-mediated disease resistance induced by pathogens, through SAR [31,37]. Among these PRs, some have a specific assigned role such as PR3 (chitinase) or PR2 (glucanase) while others, like PR1, do not have an assigned role [38]. The expression pattern of PR1 and PR3 (Figure 7) suggested that these PRs are part of the plant’s innate immunity system, and are involved in defense against this pathogen, as their expression was higher in pathogen controls. On the other hand, PR2 presented a different expression profile, showing a higher expression under all treatments, being highest on primed plants after pathogen challenge, suggesting a different role for PR2 in plant defense induced by N21.4 [22,31]. Again, the inconsistent changes in glucanase and chitinase activity and expression support the notion of pulses of gene expression driven by ROS waves [4]. Interestingly, PR2 has also been reported as a more reliable marker of priming in other species like Arabidopsis or blackberry [20] as compared to PR3 (chitinase). Since PRs are associated with SA-mediated signal transduction, it seems that N21.4 may be using at least the SA-mediated transduction pathway [5,31,39].

In summary, the present results evidence the ability of N21.4 to induce plant bio-protection, being able to prime tomato plants’ ROS scavenging enzymes and involving the enzymes PR2 and PR3 in defense mechanisms. The priming fingerprint of N21.4 is defined by an increase in growth and photochemical quenching at the physiological level, and an increase in chlorophylls, in SOD and APX activity and increased glucanase activity at the metabolic level.

## 4. Materials and Methods

### 4.1. Biological Materials: Bacterial Strains and Plants

The PGPR strain used was *P. fluorescens* N21.4, a gram-negative bacillus with a good background in inducing secondary metabolism in different plant species [21,22,23].

The leaf spot pathogen *Xanthomonas campestris* CECT 95 was used for pathogen challenge [21].

*Lycopersicum sculentum* (L.) var Razymo seeds were used, purchased from Rijk Zwaan.

### 4.2. Inoculum Preparation

Bacterial inoculum (N21.4 or pathogen) was prepared by streaking strains from −80 °C onto plate count agar (PCA), incubating at 28 °C for 24 h, and scraping bacterial cells off the plates into a volume of 10 mM MgSO_4_ to reach an absorbance value of 1 at 600 nm, corresponding to 10^8^ cfu mL^−1^. Both inocula were delivered to plants at 10^8^ cfu mL^−1^. *Pseudomonas fluorescens* was delivered to plants by soil drench and *Xanthomonas campestris* by spraying leaves with the pathogen solution [21].

### 4.3. Experimental Design

Sixteen seeds (two per pot) were placed in an eight-pot tray filled with peat (Flora Gard) and covered with vermiculite. Each tray was considered a replicate (*n* = 3) and seedlings were adjusted to 8 per tray (*n* = 8). A total of six trays (48 plants) were prepared for each treatment (N21.4 and non-inoculated controls). Controls were mock-inoculated with 10 mM MgSO_4_ and bacterial inoculation was done by soil drench upon sowing (1 mL of 10^8^ cfu mL^−1^ suspension). The second inoculation was done four weeks after sowing and, four days after, each group was divided into two subsets, one for pathogen challenge and the other for the non-pathogen controls. Pathogen was delivered to plants by leaf spray with sufficient volume to ensure that all leaves were wet, placing plants in a humid chamber 24 h before and after pathogen inoculation to ensure stomatal opening [21]. Three sampling moments were determined 24 h, 48 h and 72 h after pathogen challenge, removing 6 plants in each of them to determine enzyme activities related to oxidative stress, along with pathogenesis-related proteins (PRs) activity and gene expression by qPCR. One week after, photosynthesis was measured. Disease symptoms were recorded and results are expressed as relative disease incidence, calculated as number of diseased leaves per plant over total number of leaves, relative to pathogen control plants. Then, plants were harvested and shoot dry weight and stem length were measured (from base to oldest leaf (first leaf) and from base to youngest leaf (last leaf)); all leaves from plants in the same replicate were pooled and powdered with mortar and pestle in liquid nitrogen. This powder was stored at −80 °C and used for further analysis (chlorophylls, enzyme activities and genetic markers).

The whole experiment was done in a growth chamber (18 h/6 h light/dark period, 28 °C/20 °C). Throughout the experiment, substrate was kept moist with water and once a week half strength Hoagland solution (Phytotechlab H353) was added.

### 4.4. Chlorophyll Content Determination

To measure chlorophyll content, 0.5 g of leaf powder per replicate were mixed with 5 mL acetone (80%) by vortexing for 1 min. After centrifugation for 5 min at 4000 rpm, extracts were filtered, and absorbance was measured at 649 nm and 665 nm. Total chlorophyll and chlorophyll a and b contents were calculated [40] and expressed as nmol g^−1^ leaf weight.

### 4.5. Measurements of Chlorophyll Fluorescence

Chlorophyll fluorescence was measured with a pulse amplitude modulated (PAM) fluorometer (Hansatech FM2, Hansatech, Inc., King’s Lynn, UK) on dark-adapted leaves. The minimal fluorescence (Fo; dark adapted minimum fluorescence) was measured with weak modulated irradiation (1 μmol m^−2^ s^−1^). Maximum fluorescence (Fm) was determined for the dark-adapted state by applying a 700 ms saturating flash (9000 μmol m^−2^ s^−1^). The difference between the maximum fluorescence (Fm) and the minimum fluorescence (Fo) was the variable fluorescence (Fv). The maximum photosynthetic efficiency of photosystem II (maximal PSII quantum yield) was calculated as Fv/Fm. After the first pulse, a continuous irradiation of the leaf with red-blue actinic beams (80 μmol m^−2^ s^−1^) was carried out and equilibrated for 15 s to record Fs (steady-state fluorescence signal). Fm’ (maximum fluorescence under light adapted conditions) was determined immediately after a second saturation flash (9000 μmol m^−2^ s^−1^). The following fluorescence parameters were also calculated: effective PSII quantum yield ΦPSII = (Fm’ − Fs)/Fm’ [41] and non-photochemical quenching coefficient NPQ = (Fm − Fm’)/Fm’. All measurements were carried out in 10 plants (in the three youngest fully developed leaves per plant) of each pot.

### 4.6. Enzyme Activity Determination

The ROS cycle enzyme activity was determined at three time points (24 h, 48 h, 72 h) after pathogen challenge. The following enzyme activities related to oxidative stress were determined in plant extracts: ascorbate peroxidase (APX, EC 1.11.1.11), guaiacol peroxidase (GPX, EC 1.11.1.7), glutathione reductase (GR, EC 1.6.4.2) and superoxide dismutase (SOD, EC 1.15.1.1), as described in [42], with the modifications described in [17]. Two PR activities, ß-1,3-glucanase (PR2, EC 3.2.1.6) and chitinase (PR3, EC 3.2.1.14) were also measured [43], with the modifications described in [17].

For the colorimetric measurement of enzyme activities and total protein content, approximately 0.3 g of leaf samples were resuspended in 1 mL of potassium phosphate buffer 50 mM pH 7.5, with 1 mM EDTA, 1 mM PMSF (Phenyl-methane-sulfonyl fluoride) and 5 mM sodium ascorbate and centrifuged for 10 min at 14,000 rpm. For chitinase and ß-1,3-glucanase activities, 300 mg leaf powder were suspended in 1 mL of 0.1 M sodium acetate buffer pH 5.2.

Soluble proteins extraction and enzyme activities were determined as described by [42]. A calibration curve was constructed from commercial BSA. Protein content was expressed as mg µL^−1^.

SOD activity was determined by the degree of inhibition of the reduction of photochemical NBT with presence of riboflavin in accordance with the methodology described by [44]. This activity was determined from the inhibition of the photochemical reduction of nitro-blue-tetrazolium (NBT) in the presence of riboflavin following the manufacturer specifications of the SOD activity detection kit (Wako Pure Chemical Industries Ltd., Osaka, Japan). One unit of SOD activity is defined as the amount of enzyme capable of inhibiting the reduction rate of NBT by 50% under the test conditions above.

APX activity was determined as described by [45], with some modifications [16]. One unit of APX activity is defined as the amount of enzyme that oxidizes 1 µmol min^−1^ of ascorbate under the above assay conditions.

GPX activity was determined following the methodology described by [42], with modifications [17]. One GPX unit is defined as the amount of enzyme that produces 1 mmol min^−1^ oxidized guaiacol under the above assay conditions.

GR activity was determined following the method of [46], with modifications [17]. One GR unit is defined as the amount of enzyme that oxidizes 1 mmol min^−1^ NADPH under the above assay conditions.

PPO was determined as described by [47] and modified by [48]. The enzyme activity was expressed as changes in absorbance between t0 and t1 (60 s) at 515 nm/min^−1^ mg^−1^ of protein.

In all assays, the blank consisted of the components of the reaction mixture except for the enzyme extract, which was replaced by an equal volume of assay buffer. In the case of the GR assay, an additional blank without oxidized glutathione was included in order to record other enzyme activities able to oxidize NADPH in extracts.

Activity of Defense enzymes: Activity of pathogenesis-related protein (PRs) (ß-1,3-glucanase and chitinase) was also measured. Approximately 0.3 g of leaf powder was suspended in 1 mL of 0.1 M sodium acetate buffer pH 5.2.

ß-1,3-Glucanase activity was performed following the methodology described by [43] with modifications [17]. One unit of activity ß-1,3-glucanase is defined as the amount of enzyme which produces 1 µmol min^−1^ of reducing sugar under the given assay conditions.

Chitinase activity was determined as described by [49] with modifications [17]. One unit of chitinase activity is defined as the amount of enzyme, which produces reducing sugars (1 µmol min^−1^) under the assayed conditions. The method for making colloidal chitin is described in [50].

### 4.7. RNA Extraction and Conditions for qPCR

Differential expression of selected genes, PR1, PR2 and PR3 were verified by qPCR using the RNA samples isolated from tomato leaves obtained from each treatment. Three biological replicates were independently carried out and three plants per treatment were set up for each biological replicate. Plant tissue was ground under liquid nitrogen and total RNA from tomato leaves was extracted by PureLink^TM^ RNA Micro Kit (Invitrogen, Carlsbad, CA, USA). First-strand cDNA was synthesized from 1000 ng of total RNA using iScriptcDNA synthesis Kit (Bio-Rad, Hercules, CA, USA), according to the manufacturer’s instructions. Real-time PCR was performed using SYBR Green detection, iTaq Universal SYBR Green Supermix (Bio-Rad) on a MiniOpticon Real-Time PCR system (Bio-Rad). The primers sequence used were PR1 sense, 5′-GCCAAGCTATAACTACGCTACCAAC-3′, and antisense, 5′-GCAAGAAATGAACCACCATCC-3′; PR2 sense, 5′-GGACACCCTTCCGCTACTCTT-3′, and antisense, 5′-TGTTCCTGCCCCTCCTTTC-3′; PR3 sense, 5′-AACTATGGGCCATGTGGAAGA-3′, and antisense, 5′-GGCTTTGGGGATTGAGGAG-3′; Ubi3 was used as an endogenous reference sense, 5′-TCCATCTCGTGCTCCGTCT-3′, and antisense, 5′-GAACCTTTCCAGTGTCATCAACC-3′ (Song et al. 2010). The program used for qPCR was 3 min initial denaturation at 95 °C, followed by 40 cycles of denaturation for 20 s at 95 °C, annealing for 20 s (PR1: 51.5 °C; PR2: 51.5 °C; PR3: 58 °C; Ubi3: 51.5 °C) and extension for 20 s at 72 °C. The specificity of amplicons was verified by melting curve analysis and agarose gel electrophoresis. qPCR analyses were performed as described by [51]. Gene expression was calculated according to Ct (cycle threshold) values. For each treatment and time point, expression values of each gene (PR1, PR2 and PR3) were normalized to reference gene (actin). Next, controls were set as 1 and relative gene expression is represented in the figures. Data are expressed as relative expression to controls (set as 1.0).

### 4.8. Statistical Analysis

One-way analyses of variance with replicates were carried out to evaluate bacterial effects on all parameters When significant differences are detected (*p* < 0.05), a Fisher test was used. Prior to analysis, normality was tested by Shapiro-Wilk test (*p* > 0.05) and homoscedasticity of the variance was evaluated by Levene’s test (*p* > 0.05). The analyses were performed with the computer program Statgraphics plus 5.1 (Statistical Graphics Corp., Princeton, NJ, USA).

## 5. Conclusions

The priming fingerprint of N21.4 in the post-challenged phase is defined by bio-protection, increased growth and photochemical quenching at the physiological level; and at the metabolic level, an increase in chlorophylls, in ROS scavenging enzymes SOD, APX, GPX and GR activity and increased glucanase and chitinase activity. The bio-protection associated to increased plant growth is an encouraging feature to develop products for sustainable agriculture with this strain, as it will have an excellent performance as bio-stimulant and bio-protector.

## Figures and Tables

**Figure 1 plants-10-00331-f001:**
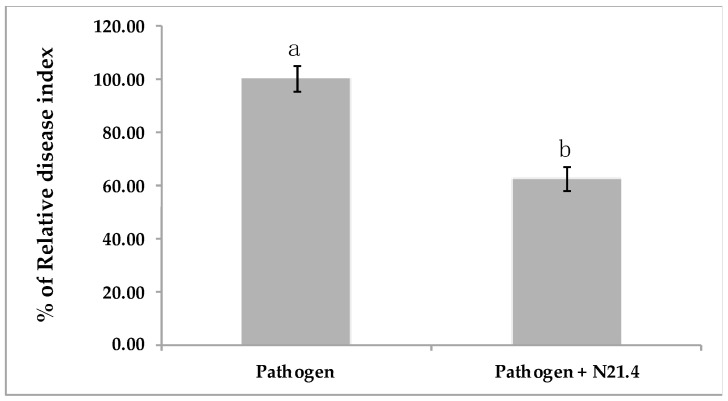
Relative disease index in 6-week-old tomato plants. Pathogen: plants challenged with the leaf spot pathogen *Xanthomonas campestris*. Pathogen + N21.4: plants treated with *Pseudomonas fluorescens* N21.4 and challenged with the leaf spot pathogen *Xanthomonas campestris*. Disease was recorded 1 week after pathogen challenge. Bars indicate standard errors (*n* = 20). Different letters denote statistically significant differences (*p* < 0.05).

**Figure 2 plants-10-00331-f002:**
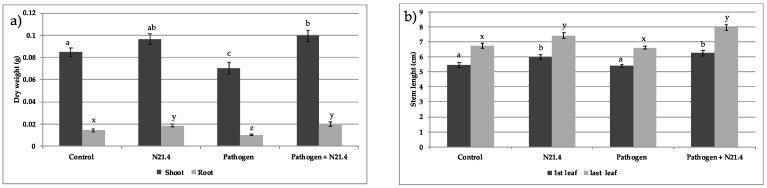
Growth parameters of 6-week old tomato plants inoculated with *P. fluorescens* N21.4 and non-inoculated controls in the priming phase and in the post-challenged priming state after inoculation with the leaf spot pathogen *Xanthomonas campestris*. (**a**) Shoot and root dry weight (g). (**b**) Stem length from base to oldest leaf (first leaf) and from base to youngest leaf (last leaf) (cm). Control: Non inoculated or challenged plants. N21.4: plants treated with *Pseudomonas fluorescens.* Pathogen: plants challenged with the leaf spot pathogen *Xanthomonas campestris*. Pathogen + N21.4: plants treated with *Pseudomonas fluorescens* N21.4 and challenged with the leaf spot pathogen *Xanthomonas campestris*. Bars indicate standard errors (*n* = 20). Different letters indicate the existence of significant differences (*p* < 0.05) in shoot (a,b,c) and roots (x,y,z) dry weight, or in stem length in the first (a,b,c) or last (x,y,z) leaf.

**Figure 3 plants-10-00331-f003:**
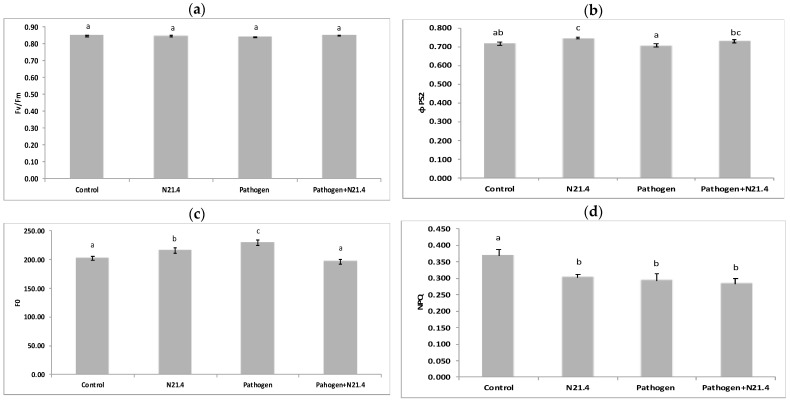
Photosynthesis parameters of 6-week old tomato plants inoculated with *P. fluorescens* N21.4 and non-inoculated controls in the priming phase and in the post-challenged priming state after inoculation with the leaf spot pathogen *Xanthomonas campestris*. (**a**) Fv/Fm (maximum quantum yield). (**b**) Φ PSII: photochemical quenching (efficiency of Photosystem II). (**c**) Fo (Minimum fluorescence). (**d**) NPQ (Non-photochemical quenching). These parameters were measured under ambient light conditions, after 1 h dark-adaptation. Control: Non treated or challenged plants. N21.4: plants treated with *Pseudomonas fluorescens.* Pathogen: plants challenged with the leaf spot pathogen *Xanthomonas campestris*. Pathogen + N21.4: plants treated with *Pseudomonas fluorescens* N21.4 and challenged with the leaf spot pathogen *Xanthomonas campestris.* Bars indicate standard errors (*n* = 12). Different letters indicate significant differences (*p* < 0.05).

**Figure 4 plants-10-00331-f004:**
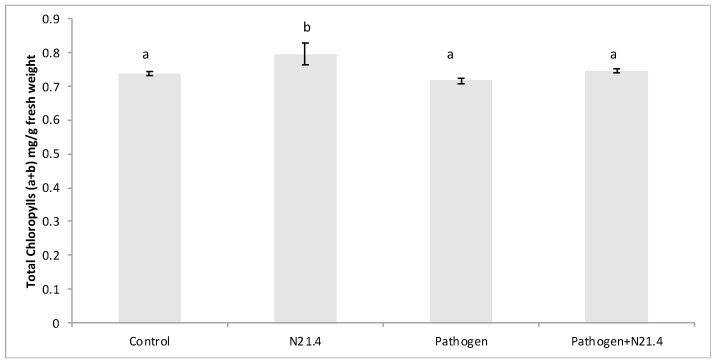
Total chlorophylls (mg of chlorophyll a + chlorophyll b per gram of fresh weight) in 6-week old tomato plants inoculated with *P. fluorescens* N21.4 and non-inoculated controls in the priming phase and in the post-challenged priming state after inoculation with the leaf spot pathogen *Xanthomonas campestris*. Control: Non treated or challenged plants. N21.4: plants treated with *Pseudomonas fluorescens.* Pathogen: plants challenged with the leaf spot pathogen *Xanthomonas campestris*. Pathogen + N21.4: plants treated with *Pseudomonas fluorescens* N21.4 and challenged with the leaf spot pathogen *Xanthomonas campestris.* Bars indicate standard errors (*n* = 3). Different letters denote significant differences (*p* < 0.05).

**Figure 5 plants-10-00331-f005:**
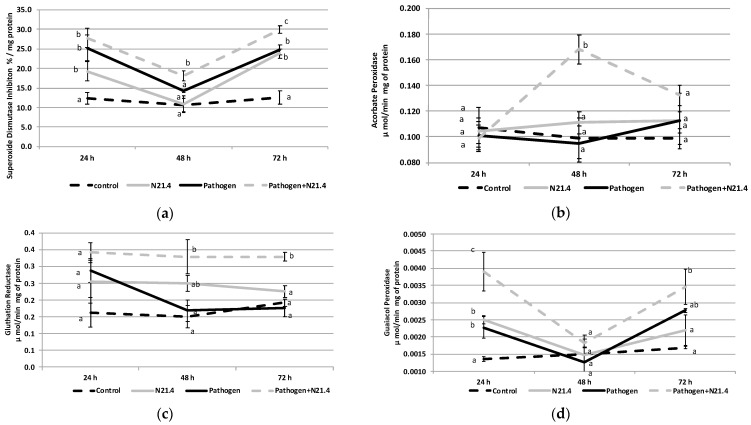
Activity of reactive oxygen species scavenging enzymes of 6-week old tomato plants inoculated with *P. fluorescens* N21.4 and non-inoculated controls in the priming phase and in the post-challenged priming state after inoculation with the leaf spot pathogen *Xanthomonas campestris* sampled at 24 h, 48 h and 72 h after challenge. (**a**) Superoxide dismutase (SOD), (**b**) Ascorbate peroxidase (APX), (**c**) Glutathione reductase (GR), (**d**) Guaiacolperoxidase (GPX), (**e**) Polyphenol oxidase (PPO). Control: Non treated or challenged plants (black broken line). N21.4: plants treated with *Pseudomonas fluorescens* (grey solid line). Pathogen: plants challenged with the leaf spot pathogen *Xanthomonas campestris* (black solid line). Pathogen + N21.4: plants treated with *Pseudomonas fluorescens* N21.4 and challenged with the leaf spot pathogen *Xanthomonas campestris* (grey broken line). Bars indicate standard errors (*n* = 3). Different letters indicate significant differences (*α* = 0.05) between treatments for each time point.

**Figure 6 plants-10-00331-f006:**
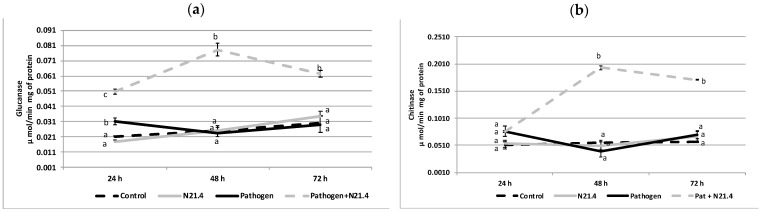
Activity of PR enzymes of 6-week old tomato plants inoculated with *P. fluorescens* N21.4 and non-inoculated controls in the priming phase and in the post-challenged priming state after inoculation with the leaf spot pathogen *Xanthomonas campestris* at 24 h, 48 h and 72 h after challenge. (**a**) ß-1,3-Glucanase (PR2). (**b**) Chitinase (PR3). Control: non treated nor challenged plants (black broken line). N21.4: plants treated with *Pseudomonas fluorescens* (grey solid line). Pathogen: plants challenged with the leaf spot pathogen *Xanthomonas campestris* (black solid line). Pathogen + N21.4: plants treated with *Pseudomonas fluorescens* N21.4 and challenged with the leaf spot pathogen *Xanthomonas campestris* (grey broken line). Bars indicate standard errors; different letters indicate significant differences (*p* < 0.05) between treatments for each time point.

**Figure 7 plants-10-00331-f007:**
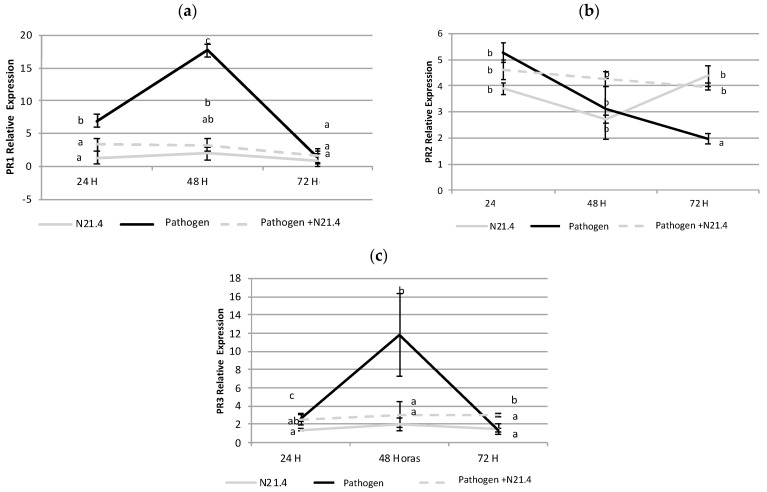
Gene expression of PR proteins (pathogenesis related proteins) of 6-week old tomato plants inoculated with *P. fluorescens* N21.4 and non-inoculated controls in the priming phase and in the post-challenged priming state after inoculation with the leaf spot pathogen *Xanthomonas campestris*, sampled at 24 h, 48 h and 72 h after challenge. (**a**) PR1, (**b**) ß-1,3-Glucanase (PR2), (**c**) Chitinase (PR3). N21.4: plants treated with *Pseudomonas fluorescens* (grey solid line). Pathogen: plants challenged with the leaf spot pathogen *Xanthomonas campestris* (black solid line). Pathogen + N21.4: plants treated with *Pseudomonas fluorescens* N21.4 and challenged with the leaf spot pathogen *Xanthomonas campestris* (grey broken line). Bars indicate standard errors (*n* = 3). Different letters indicate significant differences (α = 0.05) between treatments for each time point.

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
