# Peer review of "Tomato Bio-Protection Induced by Pseudomonas fluorescens N21.4 Involves ROS Scavenging Enzymes and PRs, without Compromising Plant Growth"

_plants, 2021, doi:10.3390/plants10020331_

Round 1
Reviewer 1 Report
The work "Bioprotection induced by Pseudomonas fluorescens N21.4 involves ROS scavenging enzymes and PRs, triggering plant protection without compromising growth on tomato" by García Villaraco and collaborators explores the nature of the tomato priming fingerprint induced by P. fluorescens N21.4. Although assays are appropriate, there are formal aspects of the manuscript that should be addressed before it is acceptable for publication.
General comments
The title contains repeated information. With "bioprotection" it is assumed that treatments with P. fluorescens trigger plant protection, which is repeated in the middle of the sentence. I suggest modifying the title to something like "Tomato bioprotection induced by Pseudomonas fluorescens N21.4 involves ROS scavenging enzymes and PRs, triggering plant protection without compromising plant growth".
The introduction focuses on priming fingerprint and other general aspects of plant responses to biotic/abiotic stress. However, despite of the abundance of works analyzing different aspects of P. fluorescens N21.4 interactions with plants - including works from the same authors -, there is no mention to why strain N21.4 is interesting, what do previous works show and what kind of new information this work will provide. It would be appropriate to provide a background or reason either in this section or in the Materials and methods, why authors chose the pathosystem Xanthomonas-tomato.
The results section needs better organization and addition of subheadings summarizing or highlighting the most relevant result/s of each assay set. In general, I would recommend revisit the results section and rephrase it. In most cases, the only results commented refer to trends -e.g. values remain constant or fluctuate-. However, although assays were designed to compare the effect of pre-treatments with P. fluorescens N21.4, these results -the relevant ones- are barely highlighted in the text.
In general, all the figures need adjustments. Although I see that authors follow similar patterns in other works, title headings in graphs should be avoided. Titles and/or descriptions of the figures should be described exclusively in the figure legend. In many cases, titles and/or measurement units are missing from Y axes. These need to be included. Additionally, any abbreviation used in graphs should be properly defined in the legend (e.g. authors constantly refer to Xanthomonas as Pathogen or Pat in graphs, but it is not specified in the legend). Figures must be self-explanatory on their own, and one should not need to read the main text to understand what the graph is illustrating.
Please, review the text and make sure abbreviations are used only after the expanded form of the word is used once (PGPR, ROS, SOD, APX, PR, etc.). Expanded and abbreviated words in the abstract should be expanded and shortened again in the main text.
Conclusions are too simplified and do not summarize the main study's achievements. This section should highlight the discovered components of the tomato priming fingerprint by P. fluorescens N21.4, effects on plant growth and also protection against X. campestris.
Particular comments
- Line 17. "ROS", change by "reactive oxygen species (ROS)" and shorten as ROS in the following sentences in the abstract. Same with SOD and APX. It is not necessary to specify "b-1,3 glucanase (PR2) and chitinase (PR3)" here, unless authors refer to these enzymes as PR2 and PR3 in the following lines in the abstract.
- Lines 41, 42. Avoid the use of kidnap/kidnapping. There are more appropriate ways to define the phenomenon explained in the text.
- Line 46. "ROS", change by "reactive oxygen species (ROS)", and refer as ROS in the remaining text. Same with other shortened words throughout the text.
- Figure 1. Transfer title to Y axis. Figure legends must be self-explanatory and every abbreviation used in the graph must be explained in the legend. E.G. In the figure legend, specify "Pseudomonas fluorescens N24.1 (N21.4)" and "Xanthomonas campestris (pathogen, pat)". I recommend avoiding the use of 2 different nomenclatures for Xanthomonas in the graph (i.e. X. campestris is named as Pathogen and Pat.). To be more precise and consistent, I would rather refer to the strains in the column titles as "Xc" and "Pf N21.4 + Xc" in all the figures.
- Figure 2. Individual graph titles should be replaced by "A" and "B" plus a proper introduction of the represented information in the figure legend. The abbreviation for grams is "g", "gr" is incorrect. Again, Y axis should contain a title and the measurement units. Please, modify. Explain in the figure legend that assays are performed on tomato plants. Again, every abbreviation used should be explained in the legend. Explain the nature of control and treatments in the figure legend. Authors specify different letters to distinguish the statistical analysis of each variable in the two graphs (a,b,c vs x,y,z), but the graphs only show a,b,c. Please, explain better what do you mean with "first" and "last" leaf, here and in the materials and methods section, and consider choosing a better self-explanatory terminology (e.g. youngest, oldest?).
- Figure 3. Y axis need a title and units. Explain which pathogen was used. Replace titles by A-D and a more appropriate explanation in the figure legend.
- Figure 4. Y axis need a title and units. Explain which pathogen was used. Replace titles by A, B, and a more appropriate explanation in the figure legend. To which extent these small (although significant) differences are biologically relevant? I have similar concerns with most figures in general.
- Figure 5. Y axis contain the measurement units, but most of them lack a title referring to the parameter which is being measured (e.g. APX activity). Define plant model, control and treatments including the name of bacterial species. The legend contains abbreviation explanations which are not used in the graphs.
- Figure 6. Same as Fig. 5.
- Figure 7. Same as Fig. 5.
- Lines 303 - 304. Specify tomato growth conditions.
- Line 308. MgSO4 is not a buffer. Change by "solution", or just remove the word "buffer". How were bacterial suspensions adjusted to 10ˆ8 cfu/mL? Please, change UFC by CFU. Please, provide details about how "inoculum was delivered to plants". At this point, one still does not know how plants were inoculated. Are authors referring to P. fluorescens? to X. campestris? Authors do not mention how the presence of P. fluorescens in 6-week-old plants was confirmed. Please, provide an explanation.
- Line 315. "MgSO4 10 mM", please change by 10 mM MgSO4.
- Line 319. Please, briefly describe the inoculation method (bacterial concentration, inoculation method, inoculated organ, etc).
- Line 325. Provide an explanation of stem length measurements including information regarding "first and last leaf"as mentioned in the corresponding figure.
- Line 328. Hoagland solution. Include a reference and/or explain composition, volume used, etc.
- Line 351. Which leaves? any criterium to choose which ones to analyze?
- Line 364. Leaf samples were resuspended. How? powdered? homogenized with mortar and pestle? Provide a more detailed description.
- Lines 374-380. SOD activity is defined here, but the represented chart shows the % inhibition, not the activity. When determining enzymatic activities, activity units are normalized per mg protein. How was the amount of protein normalized in this assay in particular?
- Line 394. A standard curve using which substrates/reagents?
- Line 435. I see authors used the Livak method for semi-quantitative gene expression analysis, but I do not understand in which moment the control treatment was set as 1. Could authors explain with some detail when this normalization was performed? before or after data analysis by the Livak method?
- Line 438. Did authors test the normal distribution of data? If so, explain which test/s were used. Significant differences can be detected, but do not "appear". Please, modify.
Author Response
Review report 1
Comments and Suggestions for Authors
The work "Bioprotection induced by Pseudomonas fluorescens N21.4 involves ROS scavenging enzymes and PRs, triggering plant protection without compromising growth on tomato" by García Villaraco and collaborators explores the nature of the tomato priming fingerprint induced by P. fluorescens N21.4. Although assays are appropriate, there are formal aspects of the manuscript that should be addressed before it is acceptable for publication.
General comments
The title contains repeated information. With "bioprotection" it is assumed that treatments with P. fluorescens trigger plant protection, which is repeated in the middle of the sentence. I suggest modifying the title to something like "Tomato bioprotection induced by Pseudomonas fluorescens N21.4 involves ROS scavenging enzymes and PRs, triggering plant protection without compromising plant growth".
Title: The title has been modified as suggested by the reviewer
The introduction focuses on priming fingerprint and other general aspects of plant responses to biotic/abiotic stress. However, despite of the abundance of works analyzing different aspects of P. fluorescens N21.4 interactions with plants - including works from the same authors -, there is no mention to why strain N21.4 is interesting, what do previous works show and what kind of new information this work will provide. It would be appropriate to provide a background or reason either in this section or in the Materials and methods, why authors chose the pathosystem Xanthomonas-tomato.
Bacterial background: We have introduced the background of P. fluorescens N21.4 in the Introduction section (lines 120-127 of revised version), highlighting the progress of knowledge represented in this work. The following paragraph has been included “Strain P. fluorescens N21.4 has been extensively studied. Its capacity to trigger plant growth and protection against leaf pathogens in Arabidopsis thaliana, tomato and soybean has been shown (21-23), as well as its ability to trigger secondary metabolism in soybean and blackberry (24,25). Furthermore, isolation of putative metabolites responsible to trigger a physiological response have been identified (26) and the metabolic pathway involved in the defensive response has been envisaged (12). However, the metabolic events underlying the systemic response in terms of ISR and antioxidant metabolism have not been studied in detail..”
Xanthomonas-tomato pathosystem: our research focuses in selecting effective strains to trigger plant physiology, being one key aspect that they are effective in different plant species. As a three-partner interaction, some specificity may happen, so on one hand, we wanted to keep the same microorganisms (beneficial and pathogen) for different experiments, changing only the plant, and on the other hand, Xanthomonas campestris. pv campestris is a is a model organism for studying plant-bacteria interaction (Qian et al., 2005, doi.10.1101/gr.3378705). Therefore, previous research was done and we conducted an evaluation to select the most effective pathogenic Xanthomonas campestris strain on A.thaliana (ref 21. Domenech et al 2007) and used the same strain for evaluating defensive response in tomato (22. Ramos-Solano et al, 2010)
We understand that this information is present in the materials and methods section as the ref. 21 is included in line 368 of the revised version. However, if you would still prefer to have this paragraph included in the ms., we will be delighted to include it.
The results section needs better organization and addition of subheadings summarizing or highlighting the most relevant result/s of each assay set. In general, I would recommend revisit the results section and rephrase it. In most cases, the only results commented refer to trends -e.g. values remain constant or fluctuate-. However, although assays were designed to compare the effect of pre-treatments with P. fluorescens N21.4, these results -the relevant ones- are barely highlighted in the text.
Results section organization and description. The results section has been reorganized and three subheadings have been included to present data grouped in physiological, metabolic and genetic changes, consistent with the objective of the work. An effort has been made to highlight relevant data for the aim of the work, especially in enzyme activity. Results have been exposed in terms of pre-challenged state (N21.4 vs. controls) and post-challenged state (N21.4+ pathogen vs. pathogen). We hope it is now clear and improved.
In general, all the figures need adjustments. Although I see that authors follow similar patterns in other works, title headings in graphs should be avoided. Titles and/or descriptions of the figures should be described exclusively in the figure legend. In many cases, titles and/or measurement units are missing from Y axes. These need to be included. Additionally, any abbreviation used in graphs should be properly defined in the legend (e.g. authors constantly refer to Xanthomonas as Pathogen or Pat in graphs, but it is not specified in the legend). Figures must be self-explanatory on their own, and one should not need to read the main text to understand what the graph is illustrating.
Figures: title headings have been eliminated and information is presented in the figure legends. Units have been included in the Y axes and abbreviations have been spelled out in legends and homogenized.
Please, review the text and make sure abbreviations are used only after the expanded form of the word is used once (PGPR, ROS, SOD, APX, PR, etc.). Expanded and abbreviated words in the abstract should be expanded and shortened again in the main text.
Abbreviations: the text has been revised to ensure that expanded and abbreviated forms are properly included in the ms.
Conclusions are too simplified and do not summarize the main study's achievements. This section should highlight the discovered components of the tomato priming fingerprint by P. fluorescens N21.4, effects on plant growth and also protection against X. campestris.
Conclusions have been modified according to reviewer’s suggestions.
Particular comments
- Line 17. "ROS", change by "reactive oxygen species (ROS)" and shorten as ROS in the following sentences in the abstract. Same with SOD and APX. It is not necessary to specify "b-1,3 glucanase (PR2) and chitinase (PR3)" here, unless authors refer to these enzymes as PR2 and PR3 in the following lines in the abstract.
These changes have been done as mentioned in the general comment above. Lines 17-18 revised version
- Lines 41, 42. Avoid the use of kidnap/kidnapping. There are more appropriate ways to define the phenomenon explained in the text.
The sentences have been rephrased to avoid the terms kidnap/kidnapping in lines 42 of the revised version.
- Line 46. "ROS", change by "reactive oxygen species (ROS)", and refer as ROS in the remaining text. Same with other shortened words throughout the text.
Revised throughout the text as suggested. Line 47 of rev.version
- Figure 1. Transfer title to Y axis. Figure legends must be self-explanatory and every abbreviation used in the graph must be explained in the legend. E.G. In the figure legend, specify "Pseudomonas fluorescens N24.1 (N21.4)" and "Xanthomonas campestris (pathogen, pat)". I recommend avoiding the use of 2 different nomenclatures for Xanthomonas in the graph (i.e. X. campestris is named as Pathogen and Pat.). To be more precise and consistent, I would rather refer to the strains in the column titles as "Xc" and "Pf N21.4 + Xc" in all the figures.
This has been addressed as indicated above. We have used the following terms consistently throughout the text: N21.4, Pathogen and Pathogen+N21.4.
- Figure 2. Individual graph titles should be replaced by "A" and "B" plus a proper introduction of the represented information in the figure legend. The abbreviation for grams is "g", "gr" is incorrect. Again, Y axis should contain a title and the measurement units. Please, modify. Explain in the figure legend that assays are performed on tomato plants. Again, every abbreviation used should be explained in the legend. Explain the nature of control and treatments in the figure legend. Authors specify different letters to distinguish the statistical analysis of each variable in the two graphs (a,b,c vs x,y,z), but the graphs only show a,b,c. Please, explain better what do you mean with "first" and "last" leaf, here and in the materials and methods section, and consider choosing a better self-explanatory terminology (e.g. youngest, oldest?).
All changes have been done as suggested. Our apologies for letters in the figure; it has been conveniently amended.
- Figure 3. Y axis need a title and units. Explain which pathogen was used. Replace titles by A-D and a more appropriate explanation in the figure legend.
Photosynthesis: as described in M&M, the analysis measures fluorescence emission which does not use units, as is the case for protocols that determine absorbance of a colorimetric reaction. All parametres are obtained from fluorescence values on dark adapted leaves after weak modulation or after a saturating pulse, plus measurements after an adaptation period.
- Figure 4. Y axis need a title and units. Explain which pathogen was used. Replace titles by A, B, and a more appropriate explanation in the figure legend. To which extent these small (although significant) differences are biologically relevant? I have similar concerns with most figures in general.
Units have been included in the Y axis of figure 4a; figure 4b does not have units as it is an index.
The pathogen used is Xanthomonas campestris as indicated in the materials and methods
Biological relevance of small significant differences in chlorophyll contents: this experiment is developed in 6 weeks, a very short period of time for tomato. The slight increase in chlorophylls under N21.4 at this time point predicts a more effective photosynthesis (significant increase in photochemical quenching PSII-fig3b), and therefore, growth (fig 2). If this plant continues growing, it will greatly benefit from this slight increase at this time point and probably, increase it making the slight difference at six weeks a large difference as time goes by.
We have realized that the chlorophyl index is redundant with the total chlorophyll content, so we have removed the chlorophyll index (fig 4b)
- Figure 5, 6, 7. Y axis contain the measurement units, but most of them lack a title referring to the parameter which is being measured (e.g. APX activity). Define plant model, control and treatments including the name of bacterial species. The legend contains abbreviation explanations which are not used in the graphs.
Titles and units have been modified and figure legends properly amended
- Lines 303 - 304. Specify tomato growth conditions.
Plant growth conditions were indicated in section “Experimental design”, L380-383 of revised version: “The whole experiment was done in a growth chamber (18h/6h light/dark period, 28°C/20ºC). Throughout the experiment, substrate was kept moist with water and once a week half strength Hoagland solution (Phytotechlab H353) was added”
- Line 308. MgSO4 is not a buffer. Change by "solution", or just remove the word "buffer".
We have removed “buffer”l353 revised version
- How were bacterial suspensions adjusted to 10ˆ8 cfu/mL?
This information has been included in the text, lines 353-356 of revised version “bacterial cells were scrapped off the plates into a volume of 10 mM MgSO4 to reach an absorbance value of 1 at 600 nm, corresponding to a solution of 10E8 cfu/mL”
- Please, change UFC by CFU.
This change has been done
- Please, provide details about how "inoculum was delivered to plants". At this point, one still does not know how plants were inoculated. Are authors referring to P. fluorescens? to X. campestris? Authors do not mention how the presence of P. fluorescens in 6-week-old plants was confirmed. Please, provide an explanation.
P.fluorescens was delivered by soil drench and X.campestris was delivered by spraying leaves with the pathogen solution; both inocula were prepared in the same way as indicated in the “inoculum preparation section” (L362-368 or revised version). We have included this information in both headings and extended the information of pathogen delivery in “experimental design section”.
The presence of N21.4 in plants was not specifically addressed. We know that, despite a non-sterile substrate is used, ours is the only microorganism massively added to plant roots, plants are kept in controlled conditions in a growth chamber and we see effects on plants.
- Line 315. "MgSO4 10 mM", please change by 10 mM MgSO4.
This modification has been done as suggested line 363 of revised version
- Line 319. Please, briefly describe the inoculation method (bacterial concentration, inoculation method, inoculated organ, etc).
We understand that this comment has been addressed in the above details.
- Line 325. Provide an explanation of stem length measurements including information regarding "first and last leaf"as mentioned in the corresponding figure.
This information has been included in the text (L476) and in figure 2 legend. “stem length from base to the older leaf (first leaf) and from the base to the youngest leaf (last leaf)”
- Line 328. Hoagland solution. Include a reference and/or explain composition, volume used, etc.
Hoagland solution can be purchased from several companies like www.phytotechlab.com ref H353. This reference has been included in the text. line 382 of revised version
- Line 351. Which leaves? any criterium to choose which ones to analyze?
We selected the 3 youngest fully developed leaves from each plant. This info has been included in the text.
- Line 364. Leaf samples were resuspended. How? powdered? homogenized with mortar and pestle? Provide a more detailed description.
After this suggestion, we realized that this info was missing in the text. It has now been included in the experimental design section as it affects all metabolic and genetic analysis. It now reads “all leaves from plants in the same replicate were pooled and powdered with mortar and pestle in liquid nitrogen. This powder was used for further analysis”
- Lines 374-380. SOD activity is defined here, but the represented chart shows the % inhibition, not the activity. When determining enzymatic activities, activity units are normalized per mg protein. How was the amount of protein normalized in this assay in particular?
We have normalized data dividing the %inhibition by mg of protein. This has been modified now
- Line 394. A standard curve using which substrates/reagents?
Concerning PPO activity, the ms.indicates that “PPO was determined as described by [45] and modified by [46]. The enzyme activity was expressed as µmol mL-1using standard curve.” The text has been revised to confirm that a standard curve was not used. The method is correct and units are changes in absorbance between t0 and t1 (60 s) at 515 nm/ min-1 mg-1 of protein
- Line 435. I see authors used the Livak method for semi-quantitative gene expression analysis, but I do not understand in which moment the control treatment was set as 1. Could authors explain with some detail when this normalization was performed? before or after data analysis by the Livak method?
Gene expression was calculated according to Ct (cycle threshold) values. For each treatment and time point, expression values of each gene (PR1, PR2 and PR3) were normalized to reference gene (actin). Next, controls were set as 1 and relative gene expression is represented in the figures. Representing controls in the figures may lead to misunderstanding so we have removed them from figure 7 and also, adjusted units to relative expression.
- Line 438. Did authors test the normal distribution of data? If so, explain which test/s were used. Significant differences can be detected, but do not "appear". Please, modify.
Normality and homocedasticity of variance were tested by Saphiro-Wilk test (p>0,05) and by Levene’s test (p>0,05), respectively. This information has been included in the test
Reviewer 2 Report
The paper Bioprotection induced by Pseudomonas fluorescens N21.4 involves ROS scavenging enzymes and PRs, triggering plant protection without compromising growth on tomato describes the effect of Pseudomonas fluorescens N21.4 on the tomato (Lycopersicum sculentum) response on Xanthomonas campestris infection. The topic of the study is interesting and the experiments were well designed and described. However, some minor issues require improvements before publication in Plants journal. For this purpose see the comments below.
Please correct scales in all figures, as decimal places should be separated by a period, not a comma. Moreover, please verify letters denoted to statistical differences in Fig. 2.
Please verify if all abbreviations are explained at their first appearance in the manuscript.
Please consider adding these papers on plant immunity and resistance that can help you enrich your article:
10.3390/plants9081020
10.3390/plants9121731
Materials and methods: The most common abbreviation for colony forming units is CFU, please correct that in the manuscript.
Line 348: was determined right after a second saturation
Line 417: It is established to use the term real-time PCR or qPCR. RT-PCR is an abbreviation of reverse transcription polymerase chain reaction.
Line 430: Please correct the reference.
Please verify carefully the whole manuscript for editorial and language issues, paying attention also to the literature section (e.g. references: 14, 18, 30, 32).
Author Response
Review report 2
Comments and Suggestions for Authors
The paper Bioprotection induced by Pseudomonas fluorescens N21.4 involves ROS scavenging enzymes and PRs, triggering plant protection without compromising growth on tomato describes the effect of Pseudomonas fluorescens N21.4 on the tomato (Lycopersicum sculentum) response on Xanthomonas campestris infection. The topic of the study is interesting and the experiments were well designed and described. However, some minor issues require improvements before publication in Plants journal. For this purpose see the comments below.
Please correct scales in all figures, as decimal places should be separated by a period, not a comma. Moreover, please verify letters denoted to statistical differences in Fig. 2.
These modifications have been addressed conveniently in all figures.
Please verify if all abbreviations are explained at their first appearance in the manuscript.
Abbreviations have been revised in the abstract and in the text to ensure that in both places are spelled out at first time and then use throughout the text.
Please consider adding these papers on plant immunity and resistance that can help you enrich your article:
10.3390/plants9081020 Martin-Rivilla, H.; Gutierrez-Mañero, F.J.; Gradillas, A.; P. Navarro, M.O.; Andrade, G.; Lucas, J.A. Identifying the Compounds of the Metabolic Elicitors of Pseudomonas fluorescens N 21.4 Responsible for Their Ability to Induce Plant Resistance. Plants 2020, 9, 1020. https://doi.org/10.3390/plants9081020
10.3390/plants9121731 Martin-Rivilla, H.; Garcia-Villaraco, A.; Ramos-Solano, B.; Gutierrez-Mañero, F.J.; Lucas, J.A. Bioeffectors as Biotechnological Tools to Boost Plant Innate Immunity: Signal Transduction Pathways Involved. Plants 2020, 9, 1731. https://doi.org/10.3390/plants9121731
Both references have been included in the text as 26 and 12 respectively
Materials and methods: The most common abbreviation for colony forming units is CFU, please correct that in the manuscript.
This typle has been corrected as suggested
Line 348: was determined right after a second saturation
This modification has been included
Line 417: It is established to use the term real-time PCR or qPCR. RT-PCR is an abbreviation of reverse transcription polymerase chain reaction.
Thank you for this comment. We have modified conveniently
Line 430: Please correct the reference.
Please verify carefully the whole manuscript for editorial and language issues, paying attention also to the literature section (e.g. references: 14, 18, 30, 32).
References have been revised and updated
Reviewer 3 Report
The present manuscript evalates the priming fingerprint of a Pseudomonas strain on tomato. The inoculum increased growth and had other beneficial effects in defence against pathogens. I have some minor comments to the authors:
Abstract: write full sentences
Fig1: How can it be more than 100%?
Fig2: grams is abreviated "g"
Conclusion or end of discussion: add comments on outlook or implementation
Author Response
Comments and Suggestions for Authors
The present manuscript evaluates the priming fingerprint of a Pseudomonas strain on tomato. The inoculum increased growth and had other beneficial effects in defence against pathogens. I have some minor comments to the authors:
Abstract: write full sentences
The abstract has been revised for proper writing. We hope it is now adequate
Fig1: How can it be more than 100%?
The Y axis reaches 120.00% but the maximum value is 100% that is the control.
Fig2: grams is abreviated "g"
This modification has been done. We apologize for the mistake.
Conclusion or end of discussion: add comments on outlook or implementation
The following comment has been included for this purpose:
“The priming fingerprint of N21.4 in the post-challenged phase is defined by bio-protection, increased growth and photochemical quenching at the physiological level; and at the metabolic level, an increase in chlorophylls, in ROS scavenging enzymes SOD, APX, GPX and GR activity and increased glucanase and chitinase activity. The bio-protection associated to increased plant growth is an encouraging feature to develop products for sustainable agriculture with this strain, as it will have an excellent perfor-mance as biostimulants and bioprotector.”
Round 2
Reviewer 1 Report
Congratulations, this is a nice work.